# A methodology for tracking cold spells in space and time: development, evaluation and applications

Weronika Osmolska<sup>1,2</sup>, Charles Chemel<sup>1,2</sup>, Amanda Maycock<sup>2</sup>, and Paul Field<sup>3</sup>

<sup>1</sup>National Centre for Atmospheric Science, University of Leeds, Leeds, UK

<sup>2</sup>School of Earth and Environment, University of Leeds, Leeds, UK

<sup>3</sup>Met Office, Exeter, UK

**Correspondence:** Weronika Osmolska (eewo@leeds.ac.uk)

Abstract. Cold spells, identified as periods of prolonged extreme low temperatures, are often analysed in an Eulerian framework or through the use of case studies. However, this restricts information about their spatio-temporal evolution and limits the ability to compare analogous events that share similar developments. This study identifies cold spells as a series of geographical objects that are connected across subsequent time-steps. These objects are characterised by persistent low-temperature anomalies and can be grouped into the same event by using a connected component method, previously applied to heatwaves. This work extends this method further by taking into account advection by the tropospheric mean wind. We also extend the methodology to filter quasi-stationary events that may have different drivers to transient events. Once catalogued, the cold spells are easily accessed based on their properties, location, or time period for further study. This study applies the cold spell identification methodology to the European Centre for ECMWF Reanalysis v5 (ERA5) dataset to develop a climatology of cold spells in the Northern hemisphere, establishing their seasonal variations and associated patterns of atmospheric circulation. We compare the results to an existing Eulerian based methodology and explain some reasons for differences in the cold spells identified. This study also demonstrates the typical pathways by which cold spells reach regions of East Asia, Europe and North America.

## 1 Introduction

Cold spells (CS) are prolonged periods of extreme low temperatures. They have been linked with exacerbated respiratory, cardiac and cerebrovascular related deaths (Keatinge et al., 1984; Donaldson et al., 1999; Chen et al., 2013). In a study analysing causes of premature deaths over the world, Gasparrini et al. (2015) found that around 5 million deaths from the 74 million analysed were linked with low temperature conditions in the 3 weeks preceding death. Physiological changes are not only related to the severity of the low temperature, but also the duration of the CS event (Bull and Morton, 1978), the relative change in temperature (Ryti et al., 2016), and the climatology – with climatologically warmer regions being at higher risk than climatologically colder regions (The Eurowinter Group, 1997). Extreme low temperatures can also cause severe ecological damage, by affecting coral reefs, mangrove forests and animal migration rates (Boucek et al., 2016), and economical damage (Sun et al., 2022).

Methods for identifying of CSs in the literature vary, but many methods use a local temperature threshold (e.g., absolute or percentile based) that must be exceeded for a specified number of consecutive days; see for instance Vavrus et al. (2006); Kolstad et al. (2010); Peings and Magnusdottir (2014); Liu et al. (2023). However, this classification does not explicitly retain information about spatio-temporal coherence of CS events, which is essential for examining their dynamics, severity, energy budget, growth rate, and pathways. To gain such insights, case studies must be conducted. Other methods of identifying CSs include extracting spatial information about temperature anomalies using self organising maps (Nygård et al., 2023), or preselecting specific regions to study anomalously low temperatures and their precursors (Huang et al., 2021; Rantanen et al., 2023). These studies are then constrained by the geographical region of choice, which can provide an incomplete representation of the extent of CSs. The need for a Lagrangian approach to examine CS dynamics has been expressed in the literature (e.g., Tuel and Martius, 2024), and whilst a spatio-temporal approach is available to study cold air mass (Iwasaki et al., 2014) and its thermodynamics, it does not investigate directly anomalous near-surface temperatures.

A method for event-based spatio-temporal tracking of temperature anomalies was developed for heatwaves, where connected geographical objects have to meet specific criteria (Lo et al., 2021). These criteria can be percentile based thresholds such as the 95th percentile of daily mean wet-bulb temperature with an absolute minimum threshold (Jackson et al., 2024). Similar connected object algorithms have been previously used to study precipitation (Sellars et al., 2015), ocean dynamics (Xue et al., 2023), storms (Dixon and Wiener, 1993), dust events (Yu et al., 2018) and floods (Debusscher and Van Coillie, 2019). However, to our knowledge such event-based method has not been applied to the study of CSs.

The present study adapts the connected object tracking algorithm and applies it to CSs. Additionally, it presents an enhanced version of the algorithm that incorporates advection, identifying regions of anomalously low temperatures that are related to the same large-scale dynamical event, but that are disjointed in space and time. The disconnection can occur because CSs exhibit different dynamics when over land, ocean, and sea regions, with varying surface and atmospheric conditions influencing their behaviour. Hence, this modified Lagrangian approach offers a more comprehensive identification of CSs compared to some other methods. Furthermore, a filter is developed to identify quasi-stationary events that might be related to slowly evolving processes (e.g., in response to persistent sea surface temperature anomalies). These new options allow the identification of CSs, based on a common set of thresholds, to be more refined.

The tracking framework used to identify CS events is described in Section 2 and compared to previous methods for characterising regional CSs in Section 3. Section 4 presents applications of this framework to CSs, while Section 5 discusses the findings. Conclusions are given in Section 6.

# 2 Methodology

35

# 2.1 Input data

The algorithm requires the following gridded input variables at a temporal resolution of 24 hours or less (see Section 2.7): 2-m temperature, zonal and meridional wind components on pressure levels for the advection option, and optionally the land-sea mask for additional filtering. For illustration purposes, the present study uses 6-hourly data, averaged to daily means, from the

Medium-Range Weather Forecasts Reanalysis v5 (ERA5) dataset (Hersbach et al., 2023a, b), gridded at 0.25° x 0.25° horizontal resolution, over the period 1940–2022, with a time interval of 6 hours, resampled to daily averages. 1940–2022.

## 2.2 Pre-processing

The 2-m temperature is deseasonalised and detrended to remove the long-term increase in temperature associated with global warming. The deseasonalisation involves Fourier transforming the climatological annual cycle of temperature at each grid cell, and retaining the six lowest frequencies. The inverse Fourier transform of these frequencies forms a smooth seasonal cycle, which is removed from each year to leave the residual temperature anomalies from the seasonal cycle. The data is then linearly detrended at each grid cell. Since this study focuses on the Northern hemisphere, the November-March extended winter temperature anomalies are extracted, to capture the coldest days of the year. A threshold is applied on temperature anomalies to extract only those that fall below the 5th percentile for 3 or more consecutive days. These are then grouped using the contouring algorithm, as described in Section 2.3.

There is an extra step in pre-processing if using the advection-based method, where a pressure-weighted average of wind fields between 800 and 250 hPa is calculated. This approach disregards surface-level winds, which are influenced by local topography, and assumes that CSs move as deep tropospheric air masses. The advection method neglects vertical motion and other processes such as diabatic heating and interaction with the surface that may affect the near-surface temperature.

# 2.3 Contouring

The contouring stage evaluates the 2-m temperature anomaly field at each time-step, masking all grid points that do not meet the threshold criterion (see Section 2.2). It then groups all contiguous grid cells that meet the criterion, including diagonal neighbours into objects. The contour of an object is defined as the outer boundary of grid cells with at least one adjacent masked neighbour and where the grid cell itself is not fully enclosed within the contour. A diagram illustrating examples of contoured objects is provided in Fig. A1. For each object, the contour and time of occurrence is identified and recorded, along with its location, temperature anomaly, wind field, and any other relevant fields inside the contour. In this study, a track is defined as a group of such contoured extreme temperature anomalies (CETAs) that can evolve coherently over time, forming a CS event. There are two ways that the algorithm can form an event, either through the overlap-based method or by the advection-based method (see the following Section 2.4).

When performing advection, the wind field is linearly interpolated to the midpoint of the grid cell edges that lie on the contour of each CETA. The wind field along the contour is then stored along with the other variables mentioned above for each CETA.

## 85 2.4 Tracking

The tracking algorithm using the overlap method follows the connectivity algorithm presented by Lo et al. (2021). This algorithm is based on whether a CETA at the next time-step overlaps with a CETA from the previous time-step. The algorithm then

Figure 1. Algorithm for tracking cold spell (CS) events. The tracks consist of contoured extreme temperature anomalies (CETAs) that meet the minimum area threshold  $A_{\min}$ . When considering the advection-based method, the CETAs are transported by the mean tropospheric wind, capped at a value set by  $U_{\max}$ . The tracks that do not contain at least one CETA of area greater than  $A_{\text{lft}}$  are discarded. QS stands for quasi-stationary filter.

performs the following logical steps (see the schematics in Fig. 1): if a CETA overlaps another CETA (by at least one grid cell) after one time-step, then this CETA is a continuation of the original CETA, and hence grouped with all its information under the same event. However, if no CETA is found to be overlapping a CETA on the previous time-step, then the CETA is regarded as decayed. The algorithm repeats this process for all CETAs. Any new CETAs that do not have a parent, from a previous time-step, will form a new event.

The tracking algorithm using advection assumes that the CS is advected by the mean tropospheric wind. CETAs that intersect with the advected contour edge at the next time-step are then considered to be part of the same event as the original CETA.






To perform this advection, the algorithm first solves a direct geodesic equation at each midpoint edge of grid cells that lie along the contour of the CETA. This uses the wind interpolated at the contour edge and the time-step duration to calculate the distance travelled, as well as the azimuthal angle from the meridional and zonal wind components to determine direction of propagation. A convex hull problem is solved by taking the starting point and all the points along the geodesic trajectory until the endpoint, from all advected contour edge points. The solution forms the smallest possible polygon that encompasses all points, which describes the advected contour. The same rules as for the overlap algorithm then apply for cataloguing the CS events, as shown in Fig. 1, except that the overlap between two contours at consecutive time-step is now determined by the intersection between the convex hull polygon and the CETAs at the next time-step.

In general, the overlap-based method can neglect discontinuities and fragmentation of CETAs from one time iteration to the next, while the advection-based method can merge distant CETAs. Figure 2 illustrates two examples in which the constituent CETAs in the CS events differ between the methods, using the well-documented winter of 2009/10 in Europe (e.g., Christiansen et al., 2018; Prior and Kendon, 2011) and East Asia, which caused heavy snow fall resulting in the closure of schools, highways, and airports (NASA Earth Observatory, 2010). Note that in these examples although the average temperature anomaly over Eurasia was low throughout the CS period, the CETAs do not cover the entire region due to the constraints in relation to the 5th percentile and consecutiveness.

In the first example (see Figs. 2a and 2b), the overlap-based method, fails to identify several CETAs belonging to the CS event, when compared to the advection-based method. The second example (see Figs. 2c and 2d), shows that the CETAs located in Europe and East Asia are grouped together when using the advection-based method, but when using the overlap-based method they are considered as two separate CS events. The advection-based algorithm is more likely to categorise two CETAs that are in proximity to each other as the same CS event. Hence in this case, the advection-based method finds that the CS occurring in Europe in early January of 2010 is related to the CS in East Asia. This approach allows one to research interconnected CS events. Furthermore, the advection-based method identifies more nearby CETAs, including those over the ocean which tend to be more discontinuous. However, both approaches show similar results for these case studies. In the first example, the CS first emerges in the Ural region and propagates westward, severely affecting the European continent. In the second example, the CS emerged also in the Ural region, but affected both north-west Europe and central Asia. The This temporal evolution was identified through a day-by-day examination of the CS track (not shown here).

To summarise, while the advection-based and overlap methods identify the same CETAs, they differ in how they group these into CS tracks. The advection-based method tracks the evolution of CETAs in space and time more coherently, capturing the

effects of winds that cause advection of cold air masses. In contrast, the overlap-based method tends to form many separate tracks of CETAs. This can be further seen by the hatched contours in Fig. 2c, which will be classed as separate CS tracks in the overlap-based method despite being part of a larger-scale event. As a result, some tracks identified by the overlap method may lack physical meaning. The differences between the two methods will be further discussed in Section 3.

# 2.5 Merging and splitting of tracks








All CS events in the catalogue are made of branches, which can merge with or split from the event. The treatment of merging and splitting of CS events in the algorithm is illustrated in the schematics in Fig. 3. Merging occurs when a CETA from an existing CS event forms a connection to a different CS event. In the example shown in Fig. 3a, two events overlap the same CETA at a given time, with the later event merged to the earlier event by incorporating it as a new branch; see Fig. 3b. Each branch within a CS event retains information about its merge location. The CETA causing the merge and its subsequent behaviour is then described within the earlier (unmerged) branch.

As a result, CS events can contain multiple branches depending on the number of merges. Merges do not always represent multiple cold air sources; more commonly, the algorithm identifies multiple branches due to fragmentation of a single CS event. For example, at the start of a CS track, CETAs can be small in scale and fragmented and therefore may be, at first, identified as separate events. In subsequent loops of the algorithm (see Fig. 1), these fragmented events are merged into a larger, more coherent cold spell, and the initial CETAs are seen as branches. Whether the CSs branches are due to multiple cold air sources or fragmentations can be easily confirmed by plotting the track of the cold spell.

Splitting occurs when a CS event overlaps with two (or more) CETAs on the next time iteration, as shown for Event 2 from time-step t+6 to time-step t+7 (see Fig. 3a). In such cases, the splittings are a continuation of the original event. There are also cases of self-merges when splitting occurs and the associated CETAs that are tracked following the splitting merge with CETAs that are part of the same event, as seen in Event 1 at t+7 (see Fig. 3a). However, a branch is only created when two separate CS events, that are currently not identified as the same event, are joined by the algorithm. Therefore, this self-merge will still be part of the same branch as it is already part of the CS event it is joining.

# 2.6 Filtering

Further criteria can be applied to filter the CS events. These criteria can be applied to the CETA themselves, filtering them before they are tracked, or to entire event as a post process. Some example criteria and the stage at which they are applied are highlighted by black rectangles in Fig. 1.

A minimum area threshold ( $A_{\min}$ ) can be set on the CETAs to reduce the tracking of small fragments that can persist and merge with other CETAs. Such CETAs are typically remnants of larger-scale events and may be considered less significant from an impact perspective. It is also recommended to apply a speed threshold  $U_{\max}$  when using the advection-based method. This threshold will prevent very high wind speeds at polar jet locations from merging CETAs together over unrealistically large distances. Furthermore, advection using very high wind speeds may not be accurate as air parcel properties can change dramatically over long distances due to other physical processes (e.g., radiation driven cooling). In this study,  $U_{\max}$  was set to

**Figure 2.** Comparison between the overlap-based and advection-based algorithms. Column 1 shows two different CS events identified through the advection-based algorithm and column 2 shows the corresponding CSs identified through the overlap-based algorithm. Filled contours show the composite area coverage of all CETAs during the event. Hatched contours in column 1 highlight CETAs that were not found through the overlap-based method. The single CS event in c) is represented by two CS events in d), shown using two different filled colors. The events a-d) occur between 07/12/2009-25/12/2009, 11/12/2009-25/12/2009, 22/12/2009-12/01/2010, and 24/12/2009-11/01/2010 and 01/01/2010-12/01/2010, respectively. Grey shading shows the average 2-m temperature anomalies during the CS days. The thin black lines show the orography, with contours every 500 m.

**Figure 3.** Illustration of how merges and splittings of CETAs are handled in the algorithm for a hypothetical CS track. a) The tracks at time-step t+9. At this time-step, CS Event 1 (green) and CS Event 2 (pink) are seen to be intersecting the same CETA. b) After this time-step, Event 2 merges into Event 1, becoming Branch 3 (green). The new Event 1 decays entirely at time-step t+11. In this case, t is an arbitary time-step in the extended winter period. Black filled circles represent the formation of a branch. Merges are shown using a bold cross, with an arrow showing the merging. Splittings are shown through a diamond shape, and a decayed CETA is represented by a cross.

 $20 \text{ m s}^{-1}$ , a value at the high-end of observed cold front speeds (Reeder and Smith, 1986; Smith and Reeder, 1988; Gohm et al., 2010). Other thresholds can also be used to filter CETAs (e.g., one can consider only the CETAs with average temperatures below freezing).

At the post-processing stage, a lifetime area threshold  $(A_{lft})$  can be applied to the CS events. This ensures that at some point during the event's lifespan, a CETA must reach a specified minimum size. This enables the study of large-scale CS events, that are likely to originate from large-scale dynamical drivers rather than from local conditions. Some large-scale events may be quasi-stationary, primarily those related to persistent temperature anomalies resulting from ocean-atmosphere interactions (e.g., La Niña events). These quasi-stationary CSs can merge with CETAs that are driven by synoptic-scale dynamical processes. A quasi-stationary filter was designed to locate cases for which in a single branch a CETA persists over any grid cell for longer than a specified period. If that branch is predominantly over the ocean (e.g., 80% of the area of the region it encompasses is over the ocean), then the branch can either be removed or kept and tagged as a different type of CS event in the catalogue. The filter then re-evaluates all branches in the affected event, amends their connections or deletes them if they no longer meet any previous filtering criteria (e.g.,  $A_{lft}$  threshold). An example of where the quasi-stationary filter is effective is demonstrated using a CS from the winter 1998/99; see Fig. 4a. This CS lasted 139 days and is predominately driven by recurring CETAs over the Pacific Ocean. These CETAs emerged over the ocean due to a particularly strong La Niña event that persisted for the entire winter (see Shabbar and Yu (2009); Dong et al. (2000)). However, additional CETAs originating from the Arctic merged with this event toward the end of their lifetime due to their proximity to the Pacific Ocean. One could interpret this as distinct events driven by different mechanisms and choose to separate them, using the quasi-stationary filter, as shown in Figs. 4b to 4d. This results in the CETAs emerging from the Arctic being considered a separate CS event.

## 175 **2.7** Sensitivity tests



The present study uses a daily time-step, and while a shorter time-step could yield more accurate results, using a significantly longer time-step is not recommended, as it would result in inaccurate representation of events. Indeed, when using the advection option, the convex hull polygon would grow progressively larger for longer time-steps, increasing the likelihood of non-physical and distant connections between CETAs. As for the overlap-based method, the percentage overlap between non-stationary CETAs would decrease with longer time-steps, potentially disjoining otherwise connected CSs.

This work focuses on large-scale extreme CS events, and thresholds are applied to extract only these events. Sensitivity tests are performed to evaluate the impact of these thresholds and ensure robustness in identifying the most significant events. To extract these CSs,  $A_{\rm lft}$  was set to be of the typical size of a mid-latitude weather system of 1,000 km across, that is  $0.8 \times 10^6$  km<sup>2</sup>. This threshold is appropriate to study CSs which likely originate from large-scale atmospheric structures, such as blocking events, rather than from local conditions (e.g., frost holes). Applying this threshold reduces the annual total cumulative area occupied by CETAs by 13% (25%) for the advection-based (overlap-based) method (see Fig. 5c). More CETAs are retained when using advection rather than overlap as the CETA overlap polygon is contained within the convex hull polygon. While both methods initially include the same CETAs,  $A_{\rm lft}$  filters out smaller events, leading to a sharp decrease in the number

Figure 4. a) Composite of CETAs from a CS event identified from 01/11/1998 to 19/03/1999 (lasting 139 days) using the advection-based algorithm, with  $A_{\min} = 70,000 \text{ km}^2$  and  $U_{\max} = 20 \text{ m s}^{-1}$  and no quasi-stationary filter applied. b-d) shows the composite CETAs belonging to the three independent CS events resulting from the event shown in a), after a quasi-stationary filter of 30 days was applied, with the quasi-stationary CS event shown in b) removed from the catalogue of events. The CSs in b-d) occur between 01/11/1998-19/03/1999, 22/11/1998-02/11/1998 and 23/01/1999-23/02/1999, respectively. The thin black lines show the orography, with contours every 500 m.

of events, by 98% (99%) for the advection-based (overlap-based) method (see Fig. 5b). As a result, the remaining events include different sets of CETAs and hence occupy different total areas.

To assess the sensitivity of  $A_{\rm min}$ , we begin by examining variations observed in the absence of other applied filters. Setting the minimum area threshold  $A_{\rm min}$  to  $10,000~{\rm km^2}$  reduces the annual average total cumulative area occupied by CETAs by 3.3%, when compared with no area threshold, for both the advection-based and overlap-based methods (see Fig. 5c). Increasing  $A_{\rm min}$  further from  $10,000~{\rm km^2}$  to  $30,000~{\rm km^2}$  and from  $30,000~{\rm km^2}$  to  $70,000~{\rm km^2}$  leads to further reductions: 3.5% and 4.5%, respectively, for the overlap-based and advection-based methods. While the area occupied by CETAs decreases steadily as  $A_{\rm min}$  increases, the number of events decreases sharply when CETAs smaller than  $10,000~{\rm km^2}$  are filtered (see Fig. 5a). The reduction is 76% for the advection-based method and 79% for the overlap-based method. Increasing  $A_{\rm min}$  further from

Figure 5. a) Number of CS tracks per year as a function of  $A_{\min}$ . Inset in a) shows a close examination of the behaviour below 40 tracks per year. b) Number of CS tracks per year as a function of  $A_{\text{lft}}$ ;  $A_{\text{lft}} = 0.8 \times 10^6 \text{ km}^2$  is highlighted. c) Total area coverage across all CETAs at various  $A_{\min}$  thresholds, averaged over all years. d) Number of CS tracks that are altered as a function of quasi-stationary persistence threshold (QS), averaged over all years; QS = 30 days is highlighted. Green lines indicate the advection-based method and red lines indicate the overlap-based method.

 $10,000 \text{ km}^2$  to  $30,000 \text{ km}^2$  and from  $30,000 \text{ km}^2$  to  $70,000 \text{ km}^2$  yields further reductions: 34% and 47%, and 28% and 39%, for the advection-based and overlap-based methods, respectively. Once  $A_{\text{lft}}$  is applied, the average number of events increases by 30% (7%) from  $A_{\text{min}}$  of  $0 \text{ km}^2$  to  $70,000 \text{ km}^2$ , for the advection-based (overlap-based) method (see Fig. 5a), as less branches are now merging from these CETAs. The advection-based method is more sensitive to this filtering, as the resulting CS events are more reliant on CETAs to merge events together. For this study,  $A_{\text{min}}$  was set to  $70,000 \text{ km}^2$ . While this value was chosen arbitrarily, it removes small-scale CETAs responsible for merges between large-scale events, with the number of events becoming more stable to slight changes in  $A_{\text{min}}$ , suggesting that the major CS events are identified.

The impact of removing quasi-stationary CETAs on the number of CS events is shown in Fig. 5d. The number of events altered (i.e., amended or removed) decreases rapidly as the threshold for persistence, the number of days a CETA must persist over a single grid point for its branch to be identified as quasi-stationary, increases. For thresholds of 10 and 30 days, 9.3% and 1.0% of events are altered for the advection-based option, and 8.7% and 0.8% for the overlap-based method, respectively.

A threshold of 30 days was chosen for this study, as a similar number of branches are filtered for both methods, suggesting that similar events are being identified. Additionally, using this threshold, 96% of the total area of the filtered branches lies within the tropics (below or at  $30^{\circ}$  latitude) for both methods. The quasi-stationary behaviour identified is likely associated with slowly evolving tropical processes rather than faster-evolving extratropical synoptic-scale weather events. Applying this threshold reduces the annual average total cumulative area occupied by CETAs by 9% for both methods, when also applying the other thresholds mentioned above. An intuitive understanding of the changes in area coverage between these filters is illustrated in Fig. 6.  $A_{\rm lft}$  and  $A_{\rm min}$  reduce the total CETAs identified per year, whereas the quasi-stationary filter reduces the CETAs identified over the tropical oceans.

Unless stated otherwise, the results presented in the remainder of this study are obtained by using the following thresholds:  $A_{\min} = 70,000 \text{ km}^2$ ,  $A_{\text{lft}} = 0.8 \times 10^6 \text{ km}^2$ , and by filtering and discarding quasi-stationary CSs longer than 30 days. This catalogue contains 2,371 and 3,171 CS events for the advection-based and overlap-based methods, respectively. It is worth noting that the methodology presented in this section is not restricted to CS events; any feature with a detectable contour and an associated wind field could be analysed using this approach. It is recommended that if the methodology is applied to a different type of event, the thresholds are either motivated by physical characteristics and behaviour of the event or selected through a similar sensitivity analysis. This ensures that the thresholds are tailored to the specific application.

#### 3 Comparison with an established method to identify cold spells





In order to assess the effectiveness of the tracking methodology, the CSs identified were compared with those following the Eulerian based method (EBM) used by Huang et al. (2021) for regions extending in longitude  $\lambda$  from  $\lambda_W$  to  $\lambda_E$  and in latitude  $\varphi$  from  $\varphi_S$  to  $\varphi_N$ . In Huang et al. (2021), CSs are identified according to the criterion:

$$CS_{\text{index}}^{t} = \frac{\int_{\varphi_{S}}^{\varphi_{N}} \int_{\lambda_{W}}^{\lambda_{E}} H\left[-\alpha \sigma\left(T_{S}^{t}\right) - T_{S}^{t}\right] \cos\left(\varphi\right) \, d\lambda \, d\varphi}{\int_{\varphi_{S}}^{\varphi_{N}} \int_{\lambda_{W}}^{\lambda_{E}} \cos\left(\varphi\right) \, d\lambda \, d\varphi} > 0.3, \quad \forall t.$$

$$(1)$$

**Figure 6.** Average number of days per year when a CS event is detected at each location using the advection-based method, when a) there is no thresholds applied, b)  $A_{\rm lft} = 0.8 \times 10^6 \ {\rm km^2}$ , c)  $A_{\rm lft} = 0.8 \times 10^6 \ {\rm km^2}$  and  $A_{\rm min} = 70,000 \ {\rm km^2}$ , d)  $A_{\rm lft} = 0.8 \times 10^6 \ {\rm km^2}$ ,  $A_{\rm min} = 70,000 \ {\rm km^2}$  and a quasi-stationary (QS) filter of 30 days. The regions defined by black polygons in a) are used for analysis in Section 3, 4.2 and 4.3. The thin black lines show the orography, with contours every 500 m.

| Region                                        | EBM, no persistence (%) | EBM (%)    | Advection (%) | Overlap (%) | $\mathbf{EBM} \cap \mathbf{Advection}\ (\%)$ | EBM ∩ Overlap (%) |
|-----------------------------------------------|-------------------------|------------|---------------|-------------|----------------------------------------------|-------------------|
| Northern Europe (55°-70°N, 0°-60°E)           | 6.2<br>4.0              | 5.0<br>3.3 | 1.9           | 1.9         | 99<br>98                                     | 99<br>98          |
| Southern Europe (35°-55°N, 0°-60°E)           | 3.9<br>2.9              | 3.1<br>2.2 | 1.3           | 1.3         | 99<br>98                                     | 99<br>98          |
| Northern America<br>(55°- 70°N, 120°- 60°W)   | 1.4<br>2.9              | 0.9<br>2.0 | 0.8           | 0.8         | 81<br>96                                     | 80<br>96          |
| Southern America<br>(35°- 55°N, 120°- 60°W)   | 3.4<br>2.6              | 2.0<br>1.5 | 0.6           | 0.5         | 99<br>96                                     | 99<br>96          |
| Northern East Asia<br>(55°-70°N, 90°-150°E)   | 2.5<br>3.0              | 1.7<br>2.2 | 1.2           | 1.2         | 99<br>100                                    | 99<br>100         |
| Southern East Asia<br>(35°- 55°N, 90°- 150°E) | 2.9<br>2.4              | 2.0<br>1.6 | 0.7           | 0.7         | 100<br>100                                   | 100<br>100        |

Table 1. Comparison of CSs found using the methodology presented in this study and the Eulerian based method (EBM) of Huang et al. (2021). The percentages represent the number of CS days identified relative to all days sampled as a fraction of 100. The intersection ( $\cap$ ) between EBM (with persistence) CS days and the tracking methods CS days is calculated as a percentage based on the total number of CS days identified by the tracking methods. The tracking methods have an additional threshold that the area of all CSs occupying the region, on a specific day, must cover at least 30% of the total area of the region. Persistence requires Eq. 1 to be valid for 3 consecutive days. Results involving the EBM are displayed in two rows, with those of the upper row based on Eq. 1 and those of the lower row based on this equation altered such that  $\alpha \sigma \left(T_S^t\right)$  is equal to the 5th percentile.

 $\operatorname{CS}^t_{\operatorname{index}}$  represents the area of CSs relative to that of the region considered. Hence, a CS is identified if it occupies 30% of the region. In this equation, H is the Heaviside function, being equal to one when  $-\alpha$   $\sigma(T_S^t) - T_S^t > 0$ , or zero otherwise. Here,  $T_S^t$  is the daily 2-m temperature anomaly  $T_S$  at time t,  $\sigma(T_S^t)$  is the local standard deviation of  $T_S$ , and  $\alpha$  is a parameter that sets the severity of the CSs. Huang et al. (2021) selected  $\alpha = 0.8$  for moderate CSs and  $\alpha = 1.2$  for severe CSs. Assuming a normal distribution the 5th percentile corresponds to  $\alpha = 1.645$ .

The same regions of study were chosen: the southern (35–55° N) and northern (55–70° N) regions of Europe (0–60° E), North America (120–60° W) and East Asia (90–150° E); these regions are illustrated in Fig. 6a. For a fair comparison with Huang et al. (2021), a constraint was placed in the tracking algorithm such that 30% of the geographical region must be covered by CETAs on a given day and an additional criterion was placed on Eq. 1 such that the CS must occur for three or more consecutive days (see Section 2.2).


The percentage of winter days where the specified criteria are met are similar for the overlap-based and advection-based methods (to within 0.1%; see Table 1), with the percentage ranging from 0.5% in South America to 1.9% in Northern Europe.

The EBM shows a significantly wider range of CS occurrence across regions, varying from 0.9% in North America to 5.0% in Northern Europe. The intersection between the tracking algorithm days and the EBM days is greater than 99% except for the

Northern America region where it is 81% and 80% when using the advection-based and overlap-based methods, respectively (see Table 1). This difference was found to result from the 2-m temperature anomalies themselves, as they did not exhibit a normal distribution, and so the value of  $\alpha$  did not correspond to the 5th percentile. To correct for this,  $\alpha$  was set to vary such that  $\alpha$   $\sigma$  ( $T_S^t$ ) is equal to the 5th percentile. The introduction of this correction led to a comparable intersection of days between the methods, greater than 95% (see Table 1). However, the EBM identifies more CS days than the tracking algorithm. There are two reasons for this: (i) the persistence criterion on regional scale used in the EBM is a weaker constraint than that used in the tracking algorithm at a grid cell scale, and (ii) the EBM does not consider the area thresholds  $A_{\min}$  and  $A_{\text{lft}}$  or the filtering of quasi-stationary CSs. Moreover, whilst the tracking algorithm does not capture all EBM CS days in a direct comparison, this does not imply that these CSs days are entirely unaccounted for. Most of these days are still represented in the catalogue, despite not meeting the 30% area threshold as seen in Fig. A2.

Overall, the intersection of days found between the methods is greater or equal to 96%, with the remaining days identified through the EBM also found in the tracking method at a smaller coverage ratio. This provides strong evidence that similar CSs are being identified using these methods. Moreover, the value of using the tracking algorithm is that a spatio-temporal reconstruction can now be be made for any CS event in the catalogue, as shown in Section 4.3.

## 4 Examples of utilisation of the catalogue for studying cold spells

# 4.1 Case studies of individual cold spell events






A CS case study is presented to highlight the information produced from the tracking algorithm. The case is that of a Eurasian cold air outbreak – causing the 5th coldest event in Japan on record (Tokyo Climate Center, 2018). The CS originated from eastern Siberia on 12/01/2018 (see Fig. 7), with the average 2-m temperature anomalies over the CETAs starting at -10 K and decreasing to as low as -19 K in the next 8 days (see Fig. 8). As the CS moved equatorward, temperature anomalies weakened. Once the cold air reached the Himalayas, the topography caused the cold air flow to diverge (see Fig. 7). Some of the cold air remained trapped in the Tarim Basin, due to its basin shaped topography, persisting from 28/01/2018 until it decayed (i.e., no longer met the tracking criteria) on 08/02/2018. This region typically experiences limited air movement due to the surrounding mountains, creating what is known as a "stagnation zone" (Wang et al., 2023). The higher concentrations of air pollution observed in this region serve as an indirect indicator of this stagnation, as the reduced airflow limits the dispersion of pollutants (Wang et al., 2019).

The cold air westward of the Himalayas propagated to Central Asia, but had a very short lifetime and decayed on 31/01/2018. In contrast the cold air that was advected eastward, propagated equatorward for longer once it passed the Himalayan region, and decayed over the warm Pacific Ocean and the Maritime Continent. A case study of this specific CS was reported by Tokyo Climate Center (2018), showing that prior to the cold air intrusion into Japan on 23/01/2018, the polar jet's position was shifted equatorward in east Siberia, thereby drawing cold air equatorward and towards Japan, as shown in Fig. 7.

**Figure 7.** Example of a CS event occurring between 12/01/2018-11/02/2018, in 3 day intervals, shown in panels a–k) respectively, using the advection-based method. Grey shading shows the 2-m temperature anomaly for each day. The blue filled contours represent the CETAs identified for each day. The dates denote the first day of each 3-day period. The thin black lines show the orography, with contours every 500 m.

**Figure 8.** 2-m temperature anomaly averaged over each CETA (points) for the CS event shown in Fig. 7. The line represents the average of these points for each day.

# 4.2 Climatology of cold spell events


The catalogue can be examined to characterise the spatio-temporal distribution of CSs over the entire Northern hemisphere. CS events are detected predominantly over land, especially over the Rocky mountains and Eurasia where they occur more than five days per year on average, with the least CSs located over the Atlantic and Pacific Oceans, as shown in Fig. 6d. CSs are also found to have preferential distributions depending on the specific month within the winter period (see Fig. 9). CSs are more prominent in northern East Asia (see Fig. 6a) in November, December and January (when 21, 23 and 26% of the CS days occur, respectively), with less CSs during February and March (18 and 12% respectively). In northern North America CSs occur predominately in December and January (when 24 and 25% of the CS days occur, respectively), with a large proportion of CSs affecting the Rocky mountains. Northern Europe experiences most CSs in January (when 29% of the CS days occur), with fewer CSs during November and March (when 12 and 13% of the CS days occur, respectively); see Table A1 for more details. Midlatitudes experience least CSs during the month of March, when fewer CSs occur. The spatial distribution of the local standard deviation of the number of CS days per month for each extended winter month, corresponding to Fig. 9, shows that year-to-year variability is largest where there are more CS events (see Fig. A3).

Figure 10 shows the direction and components of the tropospheric average wind during CS events (calculated using the wind field over the area occupied by each CETA) and of the average prevailing tropospheric wind across all winter days (climatological wind), respectively. This figure indicates that CS events in Europe are generally driven by tropospheric winds from the Norwegian Sea, the Barents Sea and Siberia (see Figs. 10a, 10c and 10e), with westward winds in Europe, in contrast to the eastward climatological winds (see Figs. 10c and 10d). In North America, the mean wind direction associated with CS events is more equatorward in the Rocky mountains and Bering Strait than that seen for the climatological winds. CSs tend

**Figure 9.** Average number of days when a CS event is identified at each location for each extended winter month with a-e) representing November to March, respectively. Results are shown for the advection-based method, however, the overlap-based method yields similar results. The thin black lines show the orography, with contours every 500 m.

**Figure 10.** Left column: average tropospheric wind from areas occupied by CS events. Right column: average climatological tropospheric wind during the extended winter period. a,b) Wind direction on a cyclical color-map. c,d) Average zonal wind speed. e,f) Average meridional wind speed. The thin black lines show the orography, with contours every 500 m.

to pass into North America through Canada, following the typical circulation pattern of the wind climatology (see Figs. 10a and 10b). However, during CS events, equatorward winds do tend to be stronger and more far reaching toward the Gulf of Mexico (see Figs. 10e and 10f). Eastward winds also appear stronger in the Pacific and Atlantic Oceans. In Greenland and Siberia, the regions of poleward wind become more localised (compared to the climatology) during CS events, occurring over a smaller geographical area (see Figs. 10a and 10b). Atmospheric conditions associated with positive Arctic Oscillation and Scandinavian blocking are known to cause a more east-equatorward flow from Greenland leading to more cold days in that region (Rantanen et al., 2023), whilst an Aleutian low can cause an equatorward flow in East Asia (Abdillah et al., 2017).

Overall, during a CS event equatorward winds intensify in southern Greenland, the Bering Sea, the Rocky Mountains, and Eurasia. In contrast, regions such as North America, southern East Asia and parts of Siberia experience equatorward tropospheric winds during CS events, that resemble those of the climatology. A monthly analysis provides a picture similar to that of the extended winter (not shown), namely there are preferential pathways for Arctic air to intrude into mid-latitudes.

### 4.3 Dynamics of regional cold spell events






Section 4.2 highlights that regions particularly affected by CSs are East Asia, Europe and North America. We now analyse the dynamics of CS events for each of these regions in turn. It is important to note that such a broad analysis was not possible before and has previously been limited to case studies. The same criteria as in Section 3 is used to illustrate this point, except that now a single event must occupy more than 30% of the defined region at some stage in its lifetime (as opposed to previously where all the CETAs in the catalogue over a single day must cover more than 30% of the defined region). This simple modification increases the sample size while allowing to focus on CSs that affect a specific region on a sufficiently large scale.

In southern East Asia (see Fig. 11), CSs first emerge in the Ural region at days -8 to -4. From day -4, the composite of CETAs shifts south-eastward. Day 0 marks the first day when CS event cover 30% of southern East Asia. The CS composite appear partially blocked by the Himalayan topography while the composite of CETAs continues to shift mostly eastward. In the next four days, CS events are transported into the Pacific Ocean and southern China, where they decay from day 2 onward. A similar picture was provided by Yao et al. (2022) for the case study of the winter 2020/21, when a pool of cold air originating from Siberia progressed equatorward; this behaviour was associated with the development of a Ural blocking event. Likewise Shoji et al. (2014) and Abdillah et al. (2017) showed that a Siberian high and Aleutian low create a pressure gradient that guides the cold air mass from the Siberian region towards East Asia. The present study shows additionally similarly shows that very large-scale CS events that occur in southern East Asia are associated with extreme low temperatures in the Ural region 8 days beforehand and on average decay 2 to 8 days after reaching southern East Asia.

In southern Europe (see Fig. 12), CSs appear over the Eastern European Plain (the Western Russian Plain), from day -6 to day -2, with some CSs also originating from the Barents Sea. The CS composite moves towards Western Europe while it is entrained in the zonal flow as it approaches the equator. The CSs typically disperse over the European continent from day 0 onwards, decaying as the CS shifts into the tropical regions and the Atlantic Ocean between day 0 to day 10. The CSs are very stationary in central Europe, but expand in area between day 0 to day 4, until the composite CS shifts toward the Atlantic Ocean after day 4. While some CSs are transported toward Eurasia throughout this period, they are small-scale, infrequent,

**Figure 11.** Subset composite of CSs events affecting East Asia and which cover at least 30% of the region shown in the black box at some stage in their lifetime, identified using the advection-based algorithm. a-i) Composite lead-lag evolution of CETAs at 2 day intervals from day -8 to 8; the days shown represent the first day of each interval. Day 0 is taken as the first day the 30% threshold is met. Vectors show the average wind direction at CETA locations on each day. Day 0 consists of 24 In total, 24 CS events were identified. The thin black lines show the orography, with contours every 500 m.

Figure 12. Same as Fig.11, but for southern Europe with a-j) showing day -6 to 12. Day 0 consists of 32 In total, 32 CS events were identified.

Figure 13. Same as Fig.11, but for southern North America region with a-h) showing day -6 to 8. Day 0 consists of 22 In total, 22 CS events were identified.

and decay quickly compared to those in Europe. The picture presented above is consistent with known CS pathways during the negative phase of the North Atlantic Oscillation (NAO-) or Scandanavian blocking in Europe (Rantanen et al., 2023).

In southern North America (see Fig. 13), CSs first appear over north-west Canada and Hudson Bay about 4 to 6 days prior to reaching the region. Between day -6 and day -2, the CS composite increases in size in central North America. From day -2 to day 2, it becomes large-scale over North America, with the Rocky mountains channeling CETAs either towards the Atlantic Ocean or equatorwards. The CSs decay over the next several days in the Atlantic Ocean, off the Rocky mountains in the Pacific ocean, and in the Gulf of Mexico. It is known that cold air surges tend to be advected along and east of the Rocky mountains toward Mexico (Colle and Mass, 1995), as shown here. A similar behaviour was found by Walsh et al. (2001), when tracking air parcels during six major CS events in North America and Europe, with cold air in North America being advected equatorward, but in Europe being advected from Siberia westward. Further research is needed to better understand the role of synoptic-scale events, such as Ural blocking or the Aleutian Low, in the development of these CS events. The algorithm presented here offers a useful tool for isolating pathways of interest, and it has been shown that these pathways are consistent with those reported in the literature.

## 4.4 Cold spell events over the Arctic Ocean








In this section a criterion is applied to retain only CS events where at least one CETA extends over the Arctic Ocean, north of 60°N. These CS events are related to marine CSs that are typically identified using sea surface temperature anomalies (Schlegel et al., 2017). A heatmap of CETAs from all the CS events that meet this criterion (see Fig. 14a) shows regions of frequent CS occurrence, namely the Bering Sea, Hudson Bay, Labrador Sea, Norwegian Sea, and the Barents and Kara Seas. All of these regions are typically associated with seasonal variations in sea-ice cover (Danielson et al., 2011; Kowal et al., 2017; Wang et al., 1994; Germe et al., 2011; Kumar et al., 2021).

Marine CSs are associated with cold air masses that move from ice covered regions to the open ocean, thereby causing an increase in sensible heat flux, potentially causing baroclinic disturbances (Mansfield, 1974), leading to polar lows. These polar lows are then associated with extreme weather conditions (Landgren et al., 2019). Regions like the Labrador, Greenland and Barents and Kara Seas have been strongly associated with marine cold air outbreaks and have been extensively studied in the literature (see for instance Kolstad et al., 2009; Narizhnaya et al., 2020; Polkova et al., 2021; Dahlke et al., 2022). While not much research is available on CSs over the Hudson Bay and Bering Sea, (Fletcher et al., 2016) indicated that the strongest marine cold air outbreaks occur over the Bering Sea. Furthermore, temperatures in the Hudson Bay have been shown to be related to the temperatures experienced in North America and Canada (Rouse, 1991; Lochte et al., 2019), with CS dynamics there being different to that over the open ocean, due to being partially enclosed by land.

To further understand the distribution of these Arctic CSs, Figs. 14b to 14f show the composite of CETAs for all the CS events that pass through the Arctic Ocean and the regions: Hudson Bay (60–70°N, 70–100°W), Labrador Sea (60–80°N, 40–70°W), Norwegian Sea (60–80°N, 40–0°W), Barents and Kara Sea (60–80°N, 10–100°E) and Bering Sea (60–70°N, 160–210°W), respectively. CSs over the Norwegian Sea, and Barents and Kara Seas, have a larger impact over Europe. Those from the Barents and Kara Seas, in particular, are typically large-scale events, predominantly influencing Central and Eastern Europe,

while extending their impact across Eurasia and parts of Western Europe. CSs from the Norwegian Sea are associated with temperature extremes occurring in the Labrador Sea and Barents and Kara Seas. These events form large-scale extreme events in north-western Europe. CSs from the Labrador and Bering Seas are typically linked to lower temperatures over the north pole. Additionally, CS events related to the Hudson Bay are connected to those from the Labrador Sea and contribute to extreme low temperatures in North America.

The application presented in this section can inform the study of future changes in the distribution of CSs over the Arctic Ocean, and their impact on continental regions.

# 5 Discussion






Section 4 demonstrates how the CS identification methodology can be used to analyse both individual and composite CS events. To date, CSs have been mostly studied using Eulerian methods, limiting the analysis of their temporal evolution, unless case studies were considered. The framework developed here allows research on CSs to be more generalised than with case studies, with the benefit of enabling the analysis of thousands of CS events on a hemispheric scale. The CSs identified through the tracking algorithm are consistent with current knowledge of CS distribution. Yet, these CS events hold the additional information about their pathways (e.g., Fig. 7, showing the CS transported from East Siberia to central Eurasia and then toward the Maritime continent). An alternative approach to this methodology would be a Lagrangian analysis of back-trajectories of air parcels relating to a CS. This would trace the pathways of air masses responsible for the CS, thereby providing a deeper understanding of its origin, dynamics and spatio-temporal distribution. However, this approach is computationally demanding, and hence making it difficult to apply to large datasets. The method used in the present work, which tracks extreme temperature anomalies from their onset point, provides a simpler means to understand the pathways of CS events and captures key dynamics without the computational complexity of a full Lagrangian trajectory analysis.

The CS tracks will be sensitive to the definition of CS. In this work, a 5th percentile threshold was used to characterise the 2-m temperature anomalies. However, a fixed percentile threshold varies significantly with geographical location. The polar climate exhibits smaller temperature variations compared to mid-latitudes. On one hand, this is advantageous, as cold air moving equatorward warms up, but can still be detected as a 5th percentile event in that region. On the other hand, it complicates the identification of the origin of the air mass. Some CSs may originate closer to the extratropics simply because they did not meet the extreme cold thresholds near the pole, even though the air mass originated from that region.

The cooling that occurs as the air mass moves toward lower latitudes is related to the rates of advective and radiative cooling, which vary depending on the location of the temperature anomaly (Nygård et al., 2023). In the Southern hemisphere, the decay of cold air mass was shown to be dominated by diabatic processes made possible through the development of mesocyclones from baroclinic instabilities. The latent heat released from the ocean was found to be a key driver of the decay, along with the sensible heat flux (Papritz and Pfahl, 2016). In the Northern hemisphere, adiabatic warming can also be an important process, depending on the trajectory of the air parcel (Nygård et al., 2023). In this study, when CSs reach warm tropical oceans (Atlantic and Pacific Oceans and Maritime Continent), they decay rapidly (see Figs. 11, 12 and 13). However, to fully understand the

Figure 14. a) Average number of days per year when a CS is present at each location and at some stage in its lifetime is located within the Arctic Ocean (north of  $60^{\circ}$ N). b-f) As in (a) but for subsets of CS events that intersect specific bounded regions: Hudson Bay, Labrador Sea, Norwegian Sea, Barents and Kara Sea, and Bering Sea, respectively (shown as white boxes in the panels). Note that no  $A_{lft}$  threshold was applied, to understand the distribution of CS all events, not just the large-scale ones. The thin black lines show the orography, with contours every 500 m.

dominant processes occurring across the CSs identified, in different regions of the Northern hemisphere, further exploration is needed regarding the respective role of adiabatic and diabatic processes.

# 6 Conclusions


This work addresses the need to characterise cold spell (CS) events, defined here as periods when the local near-surface temperature anomaly with respect to its climatological average is below the 5th percentile for a minimum of 3 days. For this purpose, a methodology is developed to identify the spatio-temporal behaviour of CSs. A novel element of the methodology is that regions of extreme temperature anomalies can be connected through space and time by advection using mean tropospheric winds. The output is a catalogue of CS events, in the form of tracks, which can be filtered for further analysis. Advection offers the benefit of identifying temperature anomalies under the same CS event that may not appear to be spatially connected at a given time. This is particularly useful over regions influenced by the same dynamical drivers but inhomogeneous surface forcing where extreme temperature anomalies can appear discontinuous.

Constraints are applied to filter the CS events, in order to reduce fragmentations or false connections, resulting in fewer CS days being identified compared to Eulerian-based methods. However, the CS events identified from the tracking algorithm allow an enhanced insights into their spatio-temporal behaviour.

The methodology is applied to the ECMWF Reanalysis v5 (ERA5) dataset, for the Northern hemisphere, over the period 1940–2022 to evaluate it and demonstrate its use for applications. The main conclusions are summarised below.

The CS events identified using this methodology are found to match those identified through the Eulerian-based approach used by Huang et al. (2021). While the Eulerian-based approach identifies a larger number of CS days based on certain thresholds, the tracking algorithm still captures the associated CS events, albeit at smaller scale. The differences in the CS days arise due to variations in the definitions of CS and the constraints applied to the CS events. Besides, the tracking algorithm provides a unique approach to retaining the spatio-temporal evolution of CSs, rather than simply identifying the days when CSs occur in a region.

Pathways for the transport of high latitude air to midlatitudes producing large-scale CSs, are concentrated in specific regions, as pointed out in the literature. CSs in East Asia are caused primarily by transport of colder air from the Ural region, with the topography playing a key role in channelling the air flow. In this case, low temperatures are detected in the Ural region 8 days before the cold air reaches southern Asia, and decays over a period of a couple of weeks. In North America, CSs are directed primarily to the east of the Rocky mountains, affecting multiple surrounding regions as they are transported toward the Gulf of Mexico and the Atlantic Ocean. However, some CSs are transported to the west of the Rocky mountains and decay in the Pacific Ocean. In Europe, CSs tend to be longer lasting when compared with the other regions and whilst the large-scale CSs originate from Siberia, other well-defined routes for the transport of colder Arctic air include along the Norwegian Sea and along the southern coast of Greenland.

The study also indicates that these preferred pathways result from anomalies in the large-scale circulation, facilitating the transport of colder Arctic air to midlatitudes. Specifically, the largest wind anomalies occur in the Bering Sea, the Rocky

mountains, Greenland and the higher latitude coastal regions of Europe and Asia. The occurrence of CSs in midlatitudes is found to vary with time within the extended winter period. CSs in the ocean at high latitudes are found to be related to anomalous temperatures over the central Arctic. Further research is required to unravel the dynamics of CSs forming near these polar coastal regions, and over the nearby oceans, to understand their impact on terrestrial regions.

The applications presented in this work can be extended to examine spatial and temporal correlations between CS events or families of CS events. The catalogue of CS events can also be stratified according to metrics describing the large-scale circulation and the state of the climate. This will be reported in a future study.

Code availability. The tracking code is available to download through GitHub (see Osmolska, 2025), and the code used to make the analysis is available upon request.

# Appendix A

**Figure A1.** Illustration of two examples of contoured extreme temperature anomalies (CETAs). Grid cells that meet the threshold criteria and are part of CETA 1 (CETA 2) are shaded in pink (green) colour. Contours are highlighted using a thick black (red) outer line for CETA 1 (CETA 2). The locations where the wind field is interpolated along the contour of CETA 1 (CETA 2) are shown using black (red) dots.

**Figure A2.** Fraction of area covered by CETAs in northern (left column) and southern (right column) Europe (first row), North America (second row) and East Asia (third row), for the set of days when the Eulerian-based method meets the 30% threshold but the advection-based tracking method does not.

**Figure A3.** Spatial distribution of the local standard deviation of the number of CS days per month for each extended winter month, corresponding to Fig. 9.

| Region                                       | Winter days (%) | November (%) | December (%) | January (%) | February (%) | March (%) |
|----------------------------------------------|-----------------|--------------|--------------|-------------|--------------|-----------|
| Northern Europe (55° - 70°N, 0° - 60°E)      | 24              | 12           | 23           | 29          | 24           | 13        |
| Southern Europe (35°- 55°N, 0°- 60°E)        | 36              | 15           | 22           | 25          | 24           | 16        |
| Northern America (55°- 70°N, 120°- 60°W)     | 24              | 21           | 24           | 25          | 17           | 13        |
| Southern America (35°- 55°N, 120°- 60°W)     | 23              | 12           | 24           | 33          | 19           | 13        |
| Northern East Asia (55° - 70°N, 90° - 150°E) | 27              | 21           | 23           | 26          | 18           | 12        |
| Southern East Asia (35° - 55°N, 90° - 150°E) | 28              | 17           | 22           | 24          | 21           | 16        |

**Table A1.** Percentage of CS days for each extended winter month by region, based on the total CS days. Here, a CS day is defined as any day when at least one CETA intersects that region.

Author contributions. WO, CC and ACM designed the study. WO performed the analysis and produced the figures under the guidance of CC and ACM. WO, CC and ACM wrote the manuscript. PF provided suggestions to improve the algorithm and comments on the manuscript.

Competing interests. The authors declare that they have no conflict of interest.

- Acknowledgements. The authors would like to thank Alan Blyth for his help with this study, as well as the two anonymous reviewers for their feedback and constructive suggestions, which helped improve this manuscript. Copernicus Climate Change Service is acknowledged for providing the ERA5 dataset. This work used JASMIN, the UK's collaborative data analysis environment (https://www.jasmin.ac.uk) for data processing (Lawrence et al., 2013). The colour palettes used to create some of the figures in this study were taken from Crameri et al. (2020).
- Financial support. This work was possible through the support from the Leeds-York-Hull Natural Environment Research Council (NERC) Doctoral Training Partnership (DTP) Panorama under grant NE/S007458/1. This work benefitted from and contributed to the Climate change in the Arctic–North Atlantic Region and Impacts on the UK (CANARI) programme funded by NERC, under grant NE/W004984/1. WO acknowledges additional funding provided by the UK Met Office through a CASE studentship. ACM was funded by the NERC StratClust project (NE/X011933/1) and the Leverhulme Trust.

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
