# Peer review of "A methodology for tracking cold spells in space and time: development, evaluation and applications"

_EGUsphere, 2025_

## Author Response (AR1)

Author response to referees' comments Osmolska et al "A methodology for tracking cold spells in space and time: development, evaluation and applications" submitted to Weather and Climate Dynamics

We thank the Editor for sourcing two constructive reviews of our manuscript. We are pleased that the reviewers find our study of value to the community and appreciate their suggestions for ways to improve the manuscript. We respond to their comments below in blue. All line numbers in our responses refer to those in the **revised track changed manuscript**.

**Referee 1**

The study introduces a new method for identifying and analyzing cold spells (CS) by treating CS as connected spatio-temporal objects rather than using traditional Eulerian frameworks or case studies. Using the ERA5 reanalysis dataset, the study develops a Northern Hemisphere climatology of cold spells, examining seasonal variations and associated atmospheric circulation patterns. The results are compared to an Eulerian-based method, highlighting differences in identified cold spells. Additionally, the study maps typical pathways of cold spells affecting East Asia, Europe, and North America.

Overall, the manuscript is well-structured and easy to follow. The methodology described in this manuscript has potential applications in risk assessment, and also sheds new sights on the understanding of the dynamics of CS in the Northern Hemisphere. Thus, I recommend this manuscript for publication pending some minor comments.

We thank the reviewer for positive feedback on our study and for suggestions to improve the clarity of the manuscript.

Question: For the advection-based and overlap-based methods, I wonder how many CSs are defined by both methods. For those events only defined by the advection-based method or only defined by the overlap-based method, what are the differences across these events? Are these related to different dynamical processes (e.g., the advection-based method captures some dynamical process that cannot be seen in the overlap-based method) or related to the differences in the methods?

**Response:** The CETAs identified using the overlap method are the same as those identified by the advection-based method, without the use of  $A_{min}$  and  $A_{lft}$ . However, the way the two methods group CETAs into CS tracks is different.

For example, as shown in Figures 2a and 2b, the overlap method can miss CETAs that are not directly interconnected in space and time. These unconnected CETAs form separate tracks, though these may not be particularly meaningful if they consist of only a single CETA. Similarly, Figures 2c and 2d show that the overlap method can sometimes split what is actually one coherent event into two separate events.

Therefore, it is not that the advection-based method identifies entirely different CETAs, but that it tracks their evolution and movement more coherently. This improved tracking may indeed reflect underlying dynamical processes more accurately, for instance capturing the effects of strong wind fields or other synoptic-scale dynamics that cause greater dispersion of cold air masses.

We have clarified this point in the revised manuscript on lines 121-126, it now reads "To summarise, while the advection-based and overlap methods identify the same CETAs, they differ in how they group these into CS tracks. The advection-based method tracks the

evolution of CETAs in space and time more coherently, capturing the effects of winds that cause advection of cold air masses. In contrast, the overlap-based method tends to form many separate tracks of CETAs. This can be further seen by the hatched contours in Fig. 2c, which will be classed as separate CS tracks in the overlap-based method despite being part of a larger-scale event. As a result, some tracks identified by the overlap method may lack physical meaning".

Question: Page 3. What does the 'resampled' mean here? Is the daily average not just simply averaging the hourly data?

**Response:** Yes, that is correct. The data was 6-hourly which was then averaged to daily average. This is changed in the revised manuscript to read '...the present study uses 6-hourly data, averaged to daily means...', on line 56.

Question: Page 3. L75. Could the authors provide a diagram here to illustrate what the contouring looks like?

Response: A figure is now included in the appendix to show the contouring and is referred to on line 76-77.

Question: Page 6. L115. How do we know that the early CS first merges in the Ural region and then propagates westward? It seems that Fig. 2 does not indicate a temporal evolution.

**Response:** This is correct. Since I had the full dataset, I could look at each day individually to determine that this occurred, but from the figure it is not possible to say this. This has been clarified in the revised version of the manuscript. Line 119-120 now reads: "This temporal evolution was identified through a day-by-day examination of the CS track (not shown here)."

Question: Page 7. L120. How do we know the merged branch is a different branch rather than a branch that was split from a previous CS event?

**Response**: If a CETA is already part of Event 1, it is simply tracked as a continuation of that event. However, if a CETA that is currently part of Event 2 overlaps or connects with Event 1, this is treated as a merging event and Event 2 is now recorded as a branch of Event 1.

On the other hand, if a CETA was previously part of Event 1 but had temporarily split, when it reconnects, it is still recognized as part of the same event. In this case, there is no new branch, because it never belonged to a different event.

A branch is only created when two separate CS events are joined by the algorithm. Therefore, a branch always represents a different CS event merging into the current one, not a returning segment from the same event.

To clarify this in the manuscript, the above sentence was implemented into line 143-145 in the paper. These lines read "However, a branch is only created when two separate CS events, that are currently not identified as the same event, are joined by the algorithm. Therefore, this self-merge will still be part of the same branch as it is already part of the CS event it is joining."

Question: Page 8. "CS events can contain multiple branches depending on the number of merges." Does this suggest that CS events have multiple cold air sources? Or the intrusion pathways?

**Response**: This depends. In some cases, though this is less common, a cold spell can result from two separate synoptic-scale events that advect cold dry air from the Arctic that eventually merge together. In this scenario, there is presence of multiple cold air sources.

However, more commonly, what the algorithm identifies as multiple branches is due to fragmentation of a single CS event. As explained in the manuscript, the algorithm may initially fail to recognize a nearby CETA as part of the same cold spell. In subsequent loops of the algorithm (see Fig. 1), these fragmented events are merged into a larger, more coherent cold spell, and the initial CETAs are seen as branches (this is explained in the next question).

Therefore, while multiple cold air sources can occasionally be involved, the branching typically reflects how the algorithm stitches together related but initially fragmented CS events, rather than indicating distinct intrusion pathways. This behaviour can be confirmed within the algorithm by plotting the track of the cold spell. Additionally, one can write code to calculate the distance travelled by each branch or visualize the atmospheric conditions at the time. These approaches can help determine whether the branch is a result of multiple cold air sources or simply from CS fragmentation.

To address this question and the question below, we have added lines 134-139 into the manuscript, that summarise our response here. Line 134-139 read "Merges do not always represent multiple cold air sources; more commonly, the algorithm identifies multiple branches due to fragmentation of a single CS event. For example, at the start of a CS track, CETAs can be small in scale and fragmented and therefore may be, at first, identified as separate events. In subsequent loops of the algorithm (see Fig. 1), these fragmented events are merged into a larger, more coherent cold spell, and the initial CETAs are seen as branches. Whether the CSs branches are due to multiple cold air sources or fragmentations can be easily confirmed by plotting the track of the cold spell."

Question: Page 8. Figure 3a. I am curious about the new branch 2 at time-step t+1. How does this branch form?

**Response:** In many cases, a new branch like Branch 2 at time-step t+1 can form due to a synoptic-scale event, such as a cold air outbreak from the Arctic. When these events first occur, they do not always produce a single, large, coherent CETA immediately. Instead, they may initially appear as multiple smaller CETAs, which gradually grow and become more organized over time.

Because these early-stage CETAs are often small in scale and may not overlap during the initial time steps, the algorithm initially identifies them as separate events. As the tracking algorithm continues looping through time (as shown in Figure 1), these CETAs can eventually merge into a single, larger event. This results in one cold spell event, but with two distinct starting points, two branches.

Question: Page 9. L155. To me, a CS event is typically a synoptic event that lasts for 1-2 weeks. How does this CS event last for 139 days?

**Response:** This event was due to a particularly strong La Niña event; see Dong et al. (2000) and Shabbar et al. (2009) for more detail. This was picked up by our algorithm due to the anomalous low temperatures in the tropical Pacific region. We therefore developed the quasi-stationary filter which enables quasi-stationary events like this to be identified and separated from cold spell events that are due to synoptic events. The original 139 day event is seen in Figure 4a, the La Niña event that was picked up by the algorithm is then separated and shown in Figure 4b, and the remaining of the event which occurs at a smaller time scale is shown in Figure 4c and 4d. The references mentioned here will be added to the manuscript to provide additional background on this La Niña event and hence the appearance of a 139 day cold spell.

These citations for the strong La Niña event have now been added to the manuscript on line 171.

**References:**

Dong, B.W., Sutton, R.T., Jewson, S.P., O'Neill, A. and Slingo, J.M., 2000. Predictable winter climate in the North Atlantic sector during the 1997–1999 ENSO cycle. Geophysical Research Letters, 27(7), pp.985-988.

Shabbar, A. and Yu, B., 2009. The 1998–2000 La Niña in the context of historically strong La Niña events. Journal of Geophysical Research: Atmospheres, 114(D13).

Question: Page 10. L185. Once  $A_{lft}$  is applied, why does the number of events increase much larger for advection-based method than for overlap-based method?

**Response**: We believe this question refers to Figure 5a inset, where after applying the  $A_{\text{lft}}$  threshold and increasing  $A_{\text{min}}$ , the number of tracks per year rises more steeply for the advection-based method compared to the overlap-based method.

This behaviour occurs because the advection algorithm creates more connections between CETAs, thereby producing fewer but longer tracks. In contrast, the overlap method forms fewer connections, leading to more fragmented tracks to begin with.

When we start removing smaller CETAs by increasing Amin, the advection-based method is more sensitive to this filtering. Since it relies more heavily on these small CETAs to link events together, removing them causes many previously connected tracks to break apart, increasing the total number of shorter tracks. The overlap method, being less reliant on these small CETAs for forming connections, shows a smaller relative increase in track count for the same threshold.

We have added the following sentence: "The advection-based method is more sensitive to this filtering, as the resulting CS events are more reliant on CETAs to merge events together." to the manuscript (line 201-202), in response to this question.

Question: Page 20. For CSs occurring over East Asia, I was wondering what percentage of these CSs originate from Ural regions. Are all of them related to a Ural blocking event? Is there a possibility that the Aleutian low can bring cold air towards East Asia?

**Response:** This is a good question! We have not investigated these events at this level of detail yet, but it offers a good avenue for future research using the dataset. We have added

this point at the end of the discussion on the dynamics of regional cold spell events, on line 336-339 in the revised manuscript. The lines read: "Further research is needed to better understand the role of synoptic-scale events, such as Ural blocking or the Aleutian Low, in the development of these CS events. The algorithm presented here offers a useful tool for isolating pathways of interest, and it has been shown that these pathways are consistent with those reported in the literature".

Question: Page 21. Figure 11. What do the authors mean "Day 0 consists of 24 events"? Does this suggest that different lags consist of different numbers of events? Same confusions about the next two plots.

**Response:** This means that 24 CS events were identified in total, and day 0 corresponds to the day each event was centred on. Because cold spells vary in duration and timing, not all events span the full time window shown in the plots. For example, by day -8, some events may not have started yet, which is why earlier (or later) days may include fewer than 24 events. This explains the variation in the number of events represented at different lags.

The manuscript now reads "In total, 24 CS events were identified". Figs. 11, 12 and 13 were changed to match this as well.

**Question: Page 21-23. Have the authors considered to show a statistical significant test here?**

Response: Since the three examples shown in Figures 11-13 are case studies for different continental regions, we do not think it adds a lot of value to show a significance test. Moreover, the figure already includes three fields ( $T_{anom}$ , wind vectors and orography) so it would make the figure too busy to show another field (significance).

**Question: Page 26. L380. I was wondering if the local wind differences across different regions have an impact on the tracking algorithm**

**Response:** The tracking algorithm is designed to be fairly robust, primarily using wind field information along the contour edges of CETAs. However, regional differences in wind speed and direction do have an impact on its behaviour:

- High wind speed regions, such as those near the polar jet, tend to produce faster
  moving CETAs. In these areas, the algorithm may attempt to link more CETAs into a
  single track due to the greater advection distances. To prevent unrealistic linkages,
  we impose a maximum wind speed threshold.
- In areas with less uniform wind, such as regions influenced by cyclonic activity, the
  wind can spread air parcels more widely. This results in larger convex hulls from the
  advected CETAs and can lead to more CETAs being linked together. However,
  because these synoptic-scale systems are large in general, this does not necessarily
  introduce significant error; the CETAs being grouped together will originate from the
  same underlying event.
- Local-scale wind variations, such as those caused by urban topography (e.g. buildings, city layouts), are not considered in the algorithm, as we are taking wind field between 800-250 hPa. This simplification helps avoid scattering in the tracking results due to small-scale variations of the wind.

Land-sea contrasts can also introduce wind differences. However, this is not
necessarily unwanted; this allows the algorithm to better reflect how air moves
differently across land and ocean surfaces.

**Referee 2**

This is a thorough evaluation of a new technique for defining cold air outbreaks. It is more a techniques paper rather than containing novel scientific results but the method and dataset produced should be useful for timely follow-up studies of tends in CAOs and how these might be affected by climate change. The work is clearly explained and well presented, and the CAO tracking algorithm/classification protocol developed should be useful as a tool for other researchers working on CAO causes and impacts.

We thank the reviewer for positive comments and are pleased that the reviewer see merit in our methodology for supporting further research of cold spells.

Question: On line 17 it is stated that 74 million deaths were analysed: how were these selected and did the selection process influence the fraction of deaths (i.e. 5 million) that were linked to low-temperature conditions

**Response:** The study referenced, by Gasparrini et al. (2015), aimed to estimate the proportion of premature deaths attributable to low and high temperatures. The authors analysed data across 13 countries: Australia, Brazil, Canada, China, Italy, Japan, South Korea, Spain, Sweden, Taiwan, Thailand, the UK, and the USA. These countries span a wide range of climates, from colder regions like Sweden to warmer ones like Brazil. The number of deaths analysed per country ranged from approximately 190,000 to over 26m and the mortality data covered multiple years (though not always the same time periods for each country). The analysis used local temperatures, and where available the air pollution was included to consider other health effects.

The analysis was based on region specific definitions of optimum temperature, and the study produced estimates of temperature attributable mortality by country, showing that cold-related deaths occurred in all regions even in warmer countries. The highest percentages of cold-related mortality were observed in China (10.36%), Italy (9.35%), Japan (9.81%) and the UK (8.48%), while the lowest were found in Thailand (2.61%), Brazil (3.53%), Sweden (3.87%) and Taiwan (4.75%).

Therefore, the selection of countries did not skew the findings toward colder regions and the diverse sample strengthened the conclusion that cold temperatures contribute to mortality despite the ambient climate.

Question: In section 2.2, line 70, it is stated that the "advection method neglects vertical motion and other processes such as diabatic heating and interaction with the surface that may affect the near-surface temperature". Have the authors checked whether this effect is negligible or would make any difference to the results?

**Response:** The advection we performed on the CETAs was based on a simple assumption: that the CETA continues to move with its average speed for one day, assuming that the anomaly persists. We then identified the expected location of the CETA at day+1 using a convex hull approach.

This can be seen as the simplest scenario for which the anomaly is advected horizontally, with no loss or modification of the air mass due to physical processes like vertical motion, diabatic heating, or surface interactions. Our intention here is not to simulate all atmospheric dynamics, but to identify a region where the cold spell is likely to be located, based purely on horizontal advection. If a new CETA appears in this region, our algorithm links the two anomalies into a single track.

As illustrated in Figure 7, this approach works reasonably well, successfully reproducing cold spells that have been previously documented in the literature and using other Eulerian based methods.

We did consider ways to account for the neglected processes. One option would be to use shorter time steps, thereby incorporating more information and potentially capturing diabatic and surface effects. However, this approach significantly increases computational demands. Another idea that we explored was limiting the advection distance, on the assumption that heating and mixing are more likely to alter the air mass over longer distances. For instance, we tested the advection using only half the projected distance, but this led to underdetection, missing CETAs that were indeed part of the same larger event.

---

## Author Response (AR2)

We thank the editor and reviewers for providing feedback on this work. Our responses to these comments are provided below.

**Reviewer's comments:**

Could the authors briefly discuss whether their methodology could be applied to examine trends in the occurrence of cold spells?

Response: The intention is to examine trends in the occurrence of cold spells. Some examples are provided in the manuscript. We are currently compiling a detailed study of past and current trends, which will be reported in a further study. In addition, we are investigating dynamical and thermodynamical drivers using clustering techniques to extract meaningful signals.

**Editor's comments:**

1. Line 5 and Line 93, 401: this summary sentence of the advection method can be further clarified by explaining the meaning of "tropospheric mean wind". Particularly, does the "mean wind" indicate daily mean or vertical mean (as in the method only horizontal wind is considered)?

Response: This is now changed, see lines 5-6.

2. Caption in Figure 2: "column 1", "column 2" should be "left column", "right column", respectively.

Response: This has now been fixed.

3. Figure 3: from my computer screen, I can't tell which lines and texts are in green color.

Response: Figure 3, green lines and text are now blue, for better visibility.

4. Paragraphs in Line 408-415, 366, 93-94, 83-84 and so on: these paragraphs only include one or two sentences. It may be better to combine them with nearby paragraphs.

Response: We have followed the recommendation.

5. Line 421: I'm wondering whether the results on Rocky mountains as the main pathways of cold spell are reliable, given that your method only use the horizontal winds. Maybe some discussions are needed.

Response: These results are robust. For the advection method presented here, the vertical component of the wind is much smaller than the horizontal component (we have clarified this in line 71). In addition, CETAs over orography would generally still be captured since the algorithm is based on two-meter temperature.